# A Dynamical Clipping Approach with Task Feedback for Proximal Policy Optimization

## Abstract

Proximal Policy Optimization (PPO) has been broadly applied to robotics learning, showcasing stable training performance. However, the fixed clipping bound setting may limit the performance of PPO. Specifically, there is no theoretical proof that the optimal clipping bound remains consistent throughout the entire training process. Meanwhile, previous researches suggest that a fixed clipping bound restricts the policy's ability to explore. Therefore, many past studies have aimed to dynamically adjust the PPO clipping bound to enhance PPO's performance. However, the objective of these approaches are not directly aligned with the objective of reinforcement learning (RL) tasks, which is to maximize the cumulative Return. Unlike previous clipping approaches, we propose a bi-level proximal policy optimization objective that can dynamically adjust the clipping bound to better reflect the preference (maximizing Return) of these RL tasks. Based on this bi-level proximal policy optimization paradigm, we introduce a new algorithm named Preference based Proximal Policy Optimization (Pb-PPO). Pb-PPO utilizes a multi-armed bandit approach to refelect RL preference, recommending the clipping bound for PPO that can maximizes the current Return. Therefore, Pb-PPO results in greater stability and improved performance compared to PPO with a fixed clipping bound. We test Pb-PPO on locomotion benchmarks across multiple environments, including Gym-Mujoco and legged-gym. Additionally, we validate Pb-PPO on customized navigation tasks. Meanwhile, we conducted comparisons with PPO using various fixed clipping bounds and various of clipping approaches. The experimental results indicate that Pb-PPO demonstrates superior training performance compared to PPO and its variants.

## 1 Introduction

Typically, there are primarily two common paradigms in reinforcement learning (RL). The first involves alternating between learning Q-networks and using them to update the policy network (Mnih et al., 2013; 2015). The second paradigm, based on gradient methods (Lillicrap et al., 2019), directly updates the policy. The second paradigm is applicable in environments with high-dimensional action space and exhibits high converge speed. But, gradient-based paradigms are generally on-policy methods, meaning that the current policy may not utilize the previously collected dataset.

To make full use of pre-collected dataset and improve algorithm's sample efficiency, we can utilize importance sampling to approximately transform on-policy algorithms into off-policy ones. However, during policy updates, a crucial challenge arises in determining the updating step size, where the new policy update may deviate too large from the old policy, compromising training stability. TRPO (Schulman et al., 2017a) addresses this concern by incorporating importance sampling and utilizing the Kullback-Leibler (KL) divergence to restrict the distance between the old policy and the new policy. This prevents excessive deviations caused by overly large updating step sizes. Subsequently, PPO (Schulman et al., 2017b) introduces a clipped surrogate objective, which is also a KL divergence term, but limits the policy update within a $\epsilon$ surrogate trust region. With this surrogate KL term, PPO achieves higher training efficiency and stability, and increasing its real-world applicability.

Specifically, PPO Schulman et al. (2017b) has been widely used in the robotics learning domain due to its stable training performance and theoretically monotonic improvement, which naturally align with the requirements of robotics learning Brohan et al. (2023a;b); Bhargava et al. (2020). However, PPO's performance is limited by the fixed setting of clipping bound. Because, a fixed clipping approach may impact the policy's exploratory Xie et al. (2022a), training stability Chen et al. (2018), and further impact the training results. Therefore, researching a better clipping approach to replace with the fixed clipping approach can be quite beneficial for further improving PPO's performance, and robotics learning.

Currently, PPO is improved from two major perspectives: 1) Modification of the advantage function (Schulman et al., 2018) to facilitate stable gradient descent or agent's exploration (Xie et al., 2022b). 2) Introducing dynamic clipping approaches to enhance the policy's exploratory capabilities (Wang et al., 2019) or ensure optimization within the boundary of optimal value (Chen et al., 2018). Regarding approach 2, most methods are replaced with dynamical clipping approach to enhance the policy's exploratory capabilities (Wang et al., 2019) or ensure the performance of optimized policy within the boundary of optimal performance (Chen et al., 2018). Regarding the modification/improvement of PPO's clipping approach, most of which are not directly aligned with the goal of RL tasks, *i.e.* maximizing expected Return, therefore, these method may not directly contribute to policy improvement.

In order to align the goal of sampling clipping bound with RL objective, we propose utilizing a multi-armed bandits approach (an almost parameter-free algorithm) and regarding PPO's evaluated Return as the RL task's feedback to dynamically recommend the clpping bound that can bring the highest return during the training process of PPO. Meanwhile, in order to improve the exploitation of candidate clipping bounds, we utilize the Upper Confidence Bound (UCB) value to guide PPO in exploring and exploiting the optimal clipping bounds throughout the entire online training stages. Our approach have several advantages over PPO and various PPO variants: 1) Compared to past clipping approaches, Pb-PPO's method of adjusting the clipping bound aligns directly with RL preference. Therefore, adjusting the clipping bound in Pb-PPO can directly enhance PPO's performance. 2) Through extensive robot experiments, we discovered that Pb-PPO's advantages over PPO extend not only to the training process but also to the model testing phase. Specifically, policies trained with Pb-PPO exhibit a more stable response to given instructions in robot tasks, showcasing the stability of Pb-PPO. To summarize, our contributions can be outlined as follows (**We recognize that while some modification of PPO have achieved competitive performance: Tao et al.; Xu et al.; Song et al. etc., our modification differs from these methods in that we focus on the clipping approach. Therefore, PPO could be further improved by combining these approaches with our method, and our majority baselines are previous clipping approaches.**):

- We propose an algorithm called Pb-PPO, which dynamically adjusts the PPO clipping bound to align with RL preference. Compared to PPO and PPO variants, Pb-PPO demonstrates more stable training performance and achieves better training results.

- We test Pb-PPO in quadruped simulation environments designed for robot deployment and find that policies trained with Pb-PPO exhibit a more stable response to given instructions. This indicates that the improvements brought by Pb-PPO also include increased stability in policies.

## 2 Related Work

**Proximal Policy Optimization (PPO).** Originally designed as an online algorithm, Proximal Policy Optimization (PPO) (Schulman et al., 2017b) aimed to enhance the applicability of Trust Region Policy Optimization (TRPO) (Schulman et al., 2017a). PPO introduced a clipping approach for efficient training, making it widely applicable across various domains, especially in robotics learning (Hoeller et al., 2023; Jenelten et al., 2024). Notably, PPO has recently been extended to the offline Reinforcement Learning (RL) setting (Zhuang et al., 2023) and multi-agent systems (Yu et al., 2022). Our study focuses on augmenting PPO, emphasizing improvements in training stability and performance within online RL settings. Recent improvements related to PPO modifications include 1) enhancing the estimation of the advantage function to ensure stable gradient descent (Schulman et al., 2018) or agent's exploration (Xie et al., 2022a), and 2) introducing adaptive clipping approaches (Chen et al., 2018) to optimize policy within the boundary of its

optimal performance and introducing dynamical clipping approaches (Wang et al., 2019) to address the limitation of fixed clipping bound that suffer from limit exploration. AAdditionally, PPO has been improved from other perspectives such as gradient updating, divergence, and exploration Tao et al. (2024); Xu et al. (2023); Song et al. (2020).

**Preference Based RL (PbRL).** Preference-based Reinforcement Learning (PbRL) is an approach used for learning from preference or feedback, extensively employed to capture and reflect human preference across diverse domains (Arumugam et al., 2019; Christiano et al., 2023; Ibarz et al., 2018; Warnell et al., 2018; Lee et al., 2021; Ouyang et al., 2022). Typically, the majority of approaches involve pre-training a reward model to reflect human preference, followed by optimization based on this pre-trained reward model, yielding notable improvements, such as enhancing Large Language Models (LLM) (Ouyang et al., 2022). Apart from learning from human feedback, PbRL also encompasses learning from task feedback in domains like Natural Language Processing (NLP) or Computer Vision (CV) (Pinto et al., 2023; Liu et al., 2022). Recently, Rafailov et al. proposed direct optimization from preference showcasing strong performance in LLM optimization. On the other hand, the primary distinction between the aforementioned approaches and Pb-PPO lies in Pb-PPO utilizing RL tasks' feedback to tune RL policy hyper-parameters rather than directly adjusting the network's parameters. Therefore, our approach is closely related to Automatic Machine Learning (AutoML) (Hutter et al., 2014; Fawcett and Hoos, 2015; van Rijn and Hutter, 2018).

## 3 Preliminary

**Reinforcement Learning.** We formulate RL as a Markov Decision Process (MDP) tuple *i.e.* $\mathcal{M} := (\mathcal{A}, \mathcal{S}, r, \gamma, d_{\mathcal{M}}, p_0, \mathbf{s}_0)$. Specifically, $\mathcal{A}$ denotes action space, $\mathcal{S}$ denotes state space, $\mathbf{a} \in \mathcal{A}$ denotes action, $\mathbf{s} \in \mathcal{S}$ denotes observation (state), $r(\mathbf{s}, \mathbf{a}) : \mathcal{A} \times \mathcal{S} \to \mathbb{R}$ denotes the reward function, $\gamma \in [0, 1]$ denotes the discount factor, $d_{\mathcal{M}}(\mathbf{s}_{t+1}|\mathbf{s}_t, \mathbf{a}_t) : \mathcal{S} \times \mathcal{A} \to \mathcal{S}$ denotes the transition function (dynamics), $p_0$ denotes the distribution of initial state, and $\mathbf{s}_0 \sim p_0$ denotes the initial state. The goal of RL is to find the optimal policy $\pi^*$ that can maximize the accumulated Return *i.e.* $\pi^* := \arg\max_\pi \mathbb{E}_{\tau \sim \pi(\tau)}[R(\tau)]$, where $\tau := \{\mathbf{s}_0, \mathbf{a}_0, \cdots, \mathbf{s}_t, \mathbf{a}_t, \cdots \mathbf{s}_T, \mathbf{a}_T | \mathbf{s}_{t+1} \sim d_{\mathcal{M}}(\cdot|\mathbf{s}_t, \mathbf{a}_t), \mathbf{a}_t = \pi(\cdot|\mathbf{s}_t), s_0 \sim p_0\}$ is the roll-out trajectory, $T$ denotes the time horizon, and $R(\tau) = \sum_{t=0}^{t=T} \gamma^t r(\mathbf{s}_t, \mathbf{a}_t)$ denotes the accumulated Return. In order to optimize policy to attain the maximum Return, off-policy algorithms iteratively estimate the expected Return of given state-action pairs $(\mathbf{s}_t, \mathbf{a}_t)$ by training a $Q$ network *i.e.* $Q(\mathbf{s}_t, \mathbf{a}_t) = \mathbb{E}_{(\mathbf{s}_t, \mathbf{a}_t) \sim \pi(\tau)}[\sum_{t'=0}^{t'=t} \gamma^{t'} r(\mathbf{s}_{t'}, \mathbf{a}_{t'})|\mathbf{s}_0 = \mathbf{s}_t, \mathbf{a}_0 = \mathbf{a}_t]$, and optimize the policy by maximizing $Q$ *i.e.* $\max_\pi \mathcal{J}(\pi) = \max_\pi \mathbb{E}_{\mathbf{s}_t \sim \pi(\tau)}[Q(\mathbf{s}_t, \pi(\cdot|\mathbf{s}_t))]$. In particular, $(\mathbf{s}_t, \mathbf{a}_t)$ can be sampled from the dataset collected across entire online training process.

Unlike off-policy algorithms, on-policy algorithms typically optimize $\pi$ using policy gradient approaches, *i.e.* $\max_\pi \mathcal{J}(\pi) = \max_\pi \mathbb{E}_{(\mathbf{s}_t, \mathbf{a}_t) \sim \pi(\tau)}(\cdot) \log \pi(\mathbf{s}_t, \mathbf{a}_t)$, on the dataset collected by the current policy. Here, $(\cdot)$ encompasses various alternatives, such as Q value $Q(\mathbf{s}_t, \mathbf{a}_t)$, advantage value $A(\mathbf{s}_t, \mathbf{a}_t) = Q(\mathbf{s}_t, \mathbf{a}_t) - V(\mathbf{s}_t)$, where $V(\mathbf{s}_t)$ is a value network used to estimate the expectation of $Q(\mathbf{s}_t, \mathbf{a}_t)$, *i.e.* $V(\mathbf{s}_t) = \mathbb{E}_{\mathbf{a} \sim \mathcal{A}}[Q(\mathbf{s}_t, \mathbf{a})]$. However, on-policy algorithms cannot utilize datasets collected from another policy *i.e.* $\pi' \in \Pi, \pi! = \pi$ to update the target policy $\pi$, which limits their sample efficiency. To enhance the on policy algorithms' efficiency of data utilization, we can introduce importance sampling *i.e.* Equation 1, approximating on-policy algorithms as off-policy algorithms. This allows us to leverage datasets collected by other policies to train the current policy.

$$\mathcal{J}_{\pi_{\text{old}}}(\pi_{\text{new}}) = \mathbb{E}_{\tau \sim \pi_{\text{old}}(\tau)}\left[\frac{\pi_{\text{new}}(\tau)}{\pi_{\text{old}}(\tau)} A^{\pi_{\text{old}}}(\mathbf{s}_t, \mathbf{a}_t)\right], \tag{1}$$

however, directly optimize Equation 1 may lead to the new policy diverging from the old policy, making it challenging to reach the optimal solution. Therefore, it's necessary to further minimize the KL divergence between the new and old policies, thus Schulman et al. propose Trust Region Optimization (TRPO) *i.e.* Equation 2.

$$\mathcal{J}_{\pi_{\text{old}}} = \mathbb{E}_{\tau \sim \pi_{\text{old}}(\tau)}\left[\frac{\pi_{\text{new}}(\tau)}{\pi_{\text{old}}(\tau)} A^{\pi_{\text{old}}}(\mathbf{s}_t, \mathbf{a}_t)\right] + D_{\text{KL}}[\pi_{\text{new}}(\cdot|\mathbf{s}_t)||\pi_{\text{old}}(\cdot|\mathbf{s}_t)]. \tag{2}$$

**Proximal Policy Optimization (PPO).** Directly computing KL divergence is computationally inefficient. Therefore, Schulman et al. proposes the surrogate trust region optimization objective called PPO, as seen in Equation 3, which enhances the computational efficiency of PPO by truncating the KL divergence within a fixed region.

$$
\mathcal{J}_{\pi_{\text{old}}}(\pi_{\text{new}}) = \mathbb{E}_{\tau \sim \pi_{\text{old}}(\tau)} \Bigg[ \min(\frac{\pi_{\text{new}}(\tau)}{\pi_{\text{old}}(\tau)} A^{\pi_{\text{old}}}(\mathbf{s}_t, \mathbf{a}_t),
$$
$$
\text{clip}(\frac{\pi_{\text{new}}(\tau)}{\pi_{\text{old}}(\tau)}, 1 - \epsilon, 1 + \epsilon) A^{\pi_{\text{old}}}(\mathbf{s}_t, \mathbf{a}_t)) \Bigg], \tag{3}
$$

where $\epsilon \in (0, 1)$ is the clipping threshold. In the section Introduction, we have initially mentioned the limitations of using fixed surrogate trust region to constrain policy updating. Therefore, we introduce Preference based PPO (Pb-PPO), which dynamically samples the clipping bound to truncate the KL divergence, and such clipping approach is aligned with the maximization of RL return. To begin with introducing Pb-PPO we first introduce multi-armed bandit and Upper Confidence Bound.

**Multi-armed bandit and Upper Confidence Bound (UCB).** Given n independent variables $\zeta = \{\epsilon_0, \epsilon_1, \cdots, \epsilon_i, \cdots, \epsilon_n\}$ with equal distribution, we treat the process of sampling these variables according to the expectation as a multi-armed bandit game. Specifically, when sample the i-th arm $\epsilon_i$, we can obtain a immediate reward $r_{t=N_{\epsilon_i}}$, where $N_{\epsilon_i}$ denotes the total times access to $\epsilon_i$, and we estimate the expected Return brought by sampling $\epsilon_i$ as $\mathbb{E}[R_{\epsilon_i}|\epsilon_i]$ *i.e.* Equation 4.

$$
U(\epsilon_i) = \mathbb{E}[R_{\epsilon_i}|\epsilon_i] = \sum_{t=0}^{t=N_{\epsilon_i}} \gamma^t r_t(\epsilon_i). \tag{4}
$$

The objective of this game is to sample the variable to achieve the highest Return. Additionally, during process of sampling from $\zeta$ and updating $\mathbb{E}[R_{\epsilon_i}|\epsilon_i]$, if we greedily sample variables (sampling with the max expected Return), this kind of sampling is termed exploitation, otherwise, it is referred to as exploration. However, if we update the estimation of expected Return without exploration (only using greedy strategy), we may fail to identify the optimal clipping bound, leading to overestimating the confidence of sub-optimal clipping bounds. To address this overestimation issue, it is crucial to strike a balance between exploitation and exploration, necessitating the introduction of uncertainty.

UCB is a strategy function defined as $U^{UCB}(\epsilon_i) = U(\epsilon_i) + \hat{U}(\epsilon_i)$ utilized to balance the exploration and exploitation of bandit arms by introducing uncertainty $\hat{U}(\epsilon_i)$, where the uncertainty value of i-th variable $\epsilon_i$ can be formulated as Equation 5. Therefore, we can sample the variable with the highest UCB value *i.e.* $\epsilon^* := \arg\max_{\epsilon_i}\{U^{UCB}(\epsilon_i)|\epsilon_i \in \zeta\}$ to efficiently balance exploration with exploitation of candidate variables.

$$
\hat{U}(\epsilon_i) = (R_{\epsilon_i}^{\max} - R_{\epsilon_i}^{\min})\sqrt{\frac{1}{2}\ln\frac{2}{\sigma}}. \tag{5}
$$

***Proof*** of Equation 5 see Appendix, where $\sigma$ denotes uncertainty factor.

# 4 Preference based Proximal Policy Optimization (Pb-PPO)

**Bi-level Proximal Policy Optimization.** We cast proximal policy optimization as a bi-level optimization problem with two primary objectives: *Objective 1):* Proximal Policy Optimization Schulman et al. (2017b), which aims to maximize the expected Return, *i.e.* $\max \mathbb{E}_{\tau \sim \pi_{\text{new}}(\tau)}[\gamma^t \cdot r(\mathbf{s}_t, \mathbf{a}_t)]$, and *Objective 2):* Updating the UCB values $\{\mathbb{E}[R_0|\epsilon_0] + \hat{U}(\epsilon_0), \cdots, \mathbb{E}[R_n|\epsilon_n] + \hat{U}(\epsilon_n)\}$ of candidate clipping bounds $\zeta = \{\epsilon_0, \epsilon_1, \cdots, \epsilon_n\}$ during the updating phases. This ensures the sampling of the optimal clipping bound, thereby optimizing PPO to reach maximum RL tasks' preference.

Specifically, In the problem setting of bi-level proximal policy optimization, $\mathbb{E}[R_{\epsilon_i}|\epsilon_i]$ represents the expected Return obtained by rolling out a policy trained with clipping bound $\epsilon_i$ *i.e.* Equation 4. Specifically, we

---

**Algorithm 1** Pb-PPO

---

**Require:** PPO modules ($\pi_{\text{new}}$, $V_\phi$, $\mathcal{D}_{\text{online}}$); candidate clipping bounds: $\zeta = \{\epsilon_0, \epsilon_1, \cdots, \epsilon_n\}$. The counter $N^{\text{bandit}}$ of total visitations, the counter $N_{\epsilon_0}^{\text{bandit}}$ of each bandit arm.

**Initialize parameters of multi-armed bandits:** tabular Return $\{\mathbb{E}[R_0|\epsilon_0] = 0, \cdots, \mathbb{E}[R_n|\epsilon_n] = 0\}$, total visitations $N^{\text{bandit}} = 0$ for all arms, and the visitation counter of each arm $N^{\text{arm}} = \{N_{\epsilon_0}^{\text{arm}} = 0, N_{\epsilon_1}^{\text{arm}} = 0, \cdots, N_{\epsilon_n}^{\text{arm}} = 0\}$, discount factor $\gamma_{bandit} \in [0, 1]$ for bandit arms, global step counter $N_{\text{step}}$

**Initialize PPO parameters:** $\pi_{\text{new}}$ and $V_\phi$

**Initialize RL hype-parameter:** Global online step as $N_{\text{step}}$, online replay buffer as $\mathcal{D}_{\text{online}}$.

**while** $N_{\text{step}} < 10^6$ **do**

    Interacting $\pi_{\text{new}}$ with environment to collect new trajectory $\tau = \{\mathbf{s}_0, \mathbf{a}_0, r_0, \cdots, \mathbf{s}_T, \mathbf{a}_T, r_T\}$, then appending $\tau$ to $\mathcal{D}_{\text{online}}$

    Update online steps $N_{\text{step}} \leftarrow N_{\text{step}} + len(\tau)$

    Alternate the new policy to old policy: $\pi_{\text{old}} \leftarrow \pi_{\text{new}}$

    Computing UCB values $\{U^{\text{UCB}}(\epsilon_i)\} = \{\mathbb{E}[R_0|\epsilon_0] + \lambda\hat{U}(\epsilon_0), \cdots, \mathbb{E}[R_n|\epsilon_n] + \lambda\hat{U}(\epsilon_n)\}$.

    Sampling a clipping bound $\epsilon_i$ according to the maximum of UCB values as the optimal clipping bound $\epsilon^*$ *i.e.* $\epsilon^* = \arg\max_{\epsilon_i}\{U^{\text{UCB}}(\epsilon_i)\}$, then utilize $\epsilon^*$ to train PPO.

    **for** j$\in$ range(updating steps) **do**

        Rewrite Equation 3 as: $\mathcal{J}_{\pi_{\text{old}}}(\pi_{\text{new}}, \epsilon^*) = \mathbb{E}_{\tau \sim \pi_{\text{old}}}[\min(\frac{\pi_{\text{new}}(\tau)}{\pi_{\text{old}}(\tau)}A^{\pi_{\text{old}}}(\mathbf{s}_t, \mathbf{a}_t), \text{clip}(\frac{\pi_{\text{new}}(\tau)}{\pi_{\text{old}}(\tau)}, 1 - \epsilon^*, 1 + \epsilon^*)A^{\pi_{\text{old}}}(\mathbf{s}_t, \mathbf{a}_t))]$

        Updating $\pi_{\text{new}}$ with $\mathcal{J}_{\pi_{\text{old}}}(\pi_{\text{new}}, \epsilon^*)$.

    **end for**

    Evaluating $\pi_{\text{new}}$ k episodes and compute un-normalized expectation $R_{\epsilon^*}^{\text{bandit}}$ for sampled clipping bound via Equation 4.

    Updating the total Return, *i.e.* $R^{\text{bandit}} \leftarrow R^{\text{bandit}} + \gamma_{bandit} * R_{\epsilon^*}^{\text{bandit}}$, update the expectation $\mathbb{E}[R_{\epsilon^i}^{\text{bandit}}|\epsilon^i]$ by $\mathbb{E}[R_{\epsilon^i}^{\text{bandit}}|\epsilon^i] = \gamma_{bandit}\mathbb{E}[R_{\epsilon^i}^{\text{bandit}}|\epsilon^i] + R_{\epsilon^i}^{\text{bandit}}$.

    Updating the total Visitation counter by $N^{\text{bandit}} \leftarrow N^{\text{bandit}} + 1$, and bandit visitation counter by $N_{\epsilon_i}^{\text{bandit}} \leftarrow N_{\epsilon_i}^{\text{bandit}} + 1$;

    Updating the normalized expectation for each bandits via Equation 8.

**end while**

---

consider the Return of the $n_\pi$-th updated policy $\pi_{\text{new}}^{n_\pi}$ when utilizing $\epsilon_n$ as reward, denoted as $r_t = r_{N_{\epsilon_i}}$. We then update the expected Return of $\epsilon_n$ using Equation 4. Consequently, we further define the bi-level objective of $n_\pi$ policy iteration of PPO as $\mathcal{J}(\pi^{n_\pi}, \zeta)$,

$$\max_{U(\epsilon^*), \mathcal{J}(\pi^{n_\pi}, \epsilon^*)} \mathcal{J}(\pi^{n_\pi}, \epsilon^*)$$
$$s.t. \ \epsilon^* \leftarrow \arg\max_{\epsilon_i}\{U^{UCB}(\epsilon_i)|\epsilon_i \in \zeta\}. \tag{6}$$

Equation 6 implies that the objective of bi-level proximal optimization is the upper bound of $\mathcal{J}(\pi^{n_\pi})$, and we can obtain the best training performance by jointly sampling the optimal clipping bound *i.e.* $\epsilon^* := \arg\max_{\epsilon_i}\{\mathcal{J}(\pi^{n_\pi}, \epsilon_i)|\epsilon_i \in \zeta\}$, and optimizing PPO with $\epsilon^*$. Subsequently, we propose Pb-PPO.

**Preference based Proximal Policy Optimization (Pb-PPO).** Pb-PPO is based on the bi-level proximal policy optimization. In addition to updating the policy through proximal policy optimization, it is also necessary, in each training epoch, to select the optimal clipping bound based on RL tasks' feedback. This ensures that the current update step length is optimized to algin with maximizing accumulated Return. In the following sections, we first introduce the implementation of *Objective 2)*. Subsequently, in section 4.2, we will provide a comprehensive overview of the implementation process of Pb-PPO.

### 4.1 Implementation of *Objective 2)*

**Notations.** We first define crucial symbols. Specifically, we define $N_{\text{step}}$ as the number of policy updating iterations, $N_{\epsilon_i}$ as the visitation counter for the bandit clipping bound (bandit arm) $\epsilon_i$, $R^{\text{bandit}}$ as the total accumulated Return, which is the discounted sum of all evaluated Returns after updating the old policy across all training iterations (illustrated in line 24 of Algorithm 1, labeled by blue), and $R_{\epsilon_i}^{\text{bandit}}$ as the discounted sum of all evaluated Returns after updating the old policy when choosing $\epsilon_i$.

**Sampling clipping bound with alternate uncertainty term.** We sample the clipping bound with the highest UCB value, *i.e.*, $\epsilon_i = \arg\max_\epsilon \{U^{\mathrm{UCB}}(\epsilon_i) = U(\epsilon_i) + \lambda \hat{U}(\epsilon_i) | \epsilon_i \in \zeta\}$. Despite Equation 5 strictly adhering to UCB theory, a concern we may encounter is the inherent fluctuation in RL training curves. If Equation 5 is directly employed, several issues may arise, including encountering local minima, especially when initially rolling out a trajectory with poor performance leading to a sustained large value for $\hat{U}$.

To address this concern, we utilize Equation 7 instead to compute the uncertainty factor. Specifically, given the sampling times $N_{\epsilon_i}^{\mathrm{bandit}}$ of $\epsilon_i$ and total sampling times $N^{\mathrm{bandit}} = \sum_{\epsilon_i \in \zeta} N_{\epsilon_i}^{\mathrm{bandit}}$. If a certain clipping bound is sampled infrequently, resulting in a lower $N_{\epsilon_i}^{\mathrm{bandit}}$, it will correspondingly yield a higher $\frac{N^{\mathrm{bandit}}}{N_{\epsilon_i}^{\mathrm{bandit}}}$, leading to a larger $U^{\mathrm{UCB}}$. This encourages the exploitation of such clipping bounds. Additionally, eps is a very small float number set to prevent value overflow.

$$\hat{U}(\epsilon_i) = \sqrt{\frac{N^{\mathrm{bandit}}}{N_{\epsilon_i}^{\mathrm{bandit}} + \mathrm{eps}}}. \tag{7}$$

**Connection between Equation 7 and Equation 5.** Equation 7 represents an modification or real implementation over Equation 5 based on some empirical analysis. From Equation 7, we can observe that for $\pi_\epsilon$ with relatively fewer samples, it will obtain a larger value of $\frac{N^{\mathrm{bandit}}}{N_{\epsilon_i}^{\mathrm{bandit}} + \mathrm{eps}}$, which further leads to a larger $\hat{U}(\epsilon_i)$. This encourages more exploration of such $\epsilon$. Meanwhile, the less explored $\epsilon$ is used less frequently, which may result in a larger variance in performance, corresponding to a larger $R_{\epsilon_i}^{\max} - R_{\epsilon_i}^{\min}$, further we will have a larger $\hat{U}(\epsilon_i)$. Therefore, it implies that $\epsilon$ that is fewer exploited will receive a larger $\hat{U}(\epsilon_i)$, which aligns with the pattern observed in Equation 7. As mentioned earlier, Equation 7 can mitigate the impact of excessive fluctuations in the RL evaluation scores from Equation 5. Therefore, we choose Equation 7 to implement Pb-PPO.

**Estimation of candidate clipping bounds' expected Return.** In this section, we discuss updating the expectation of arm $\epsilon_i$, *i.e.*, updating $\mathbb{E}[R_{\epsilon_i}|\epsilon_i]$. Specifically, for each sampled arm $\epsilon_i$, we first update the policy $\pi_{\mathrm{old}}$ with the clipping bound $\epsilon_i$. We then evaluate this updated policy $\pi_{\mathrm{new}}$ k times, and calculate the average evaluated return $R_{\epsilon_i}^{\mathrm{bandit}}$, which serves as the reward $r_{N_{\epsilon_i}}$ for arm $\epsilon_i$. Subsequently, we update the expected return of sampling $\epsilon_i$, *i.e.*, $\mathbb{E}[R_{\epsilon_i}|\epsilon_i]$, using Equation 4. Meanwhile, we can normalize the expected return of arm $\epsilon_i$ by calculating the advantage, as shown in Equation 8.

$$\mathrm{norm}(\mathbb{E}[R_{\epsilon_i}|\epsilon_i]) = \mathbb{E}[R_{\epsilon_i}|\epsilon_i] - \bar{R}^{\mathrm{bandit}}. \tag{8}$$

### 4.2 Practical Implementation of Pb-PPO

Our codebase is constructed based on `stable_baselines3` [1], and the implementation has been detailed in Algorithm 1. In our practical implementation for Gym-Mujoco domain, we compute $R_{\epsilon_i}^{\mathrm{bandit}}$ by averaging the rollout performance over k=2 or 10 iterations, achieving stable performance with the weight of the random item $\lambda$ set to 5. Additionally, we generate candidate clipping bounds by uniformly sampling from a specified range. For example, when setting up 10 bounds with the minimum clipping bound as 0 and the maximum clipping bound as 1, we obtain the following 10 candidate clipping bounds: $\{0, 0.1, 0.2, 0.4, \cdots, 1\}$. Furthermore, we propose using Equation 8 to normalize the expected Return of the sampled bound $\epsilon_i$. This equation reflects the advantage of the current sampled clipping bound compared to the average expected Return across all candidate clipping bounds. We denote this setting as *Pb-PPO (wi-ad)*. Additionally, we validate Pb-PPO without normalization, denoted as *Pb-PPO (wo-ad)*. For the settings of tasks sourced from legged-gym and AUV navigation domains, please refer to Appendix.

## 5 Evaluation

**Our benchmarks.** To compare the performance of Pb-PPO with various PPO variants, we conducted evaluations on locomotion tasks in the Gym-Mujoco domain Brockman et al. (2016). Additionally, to validate

---

[1] https://github.com/DLR-RM/stable-baselines3

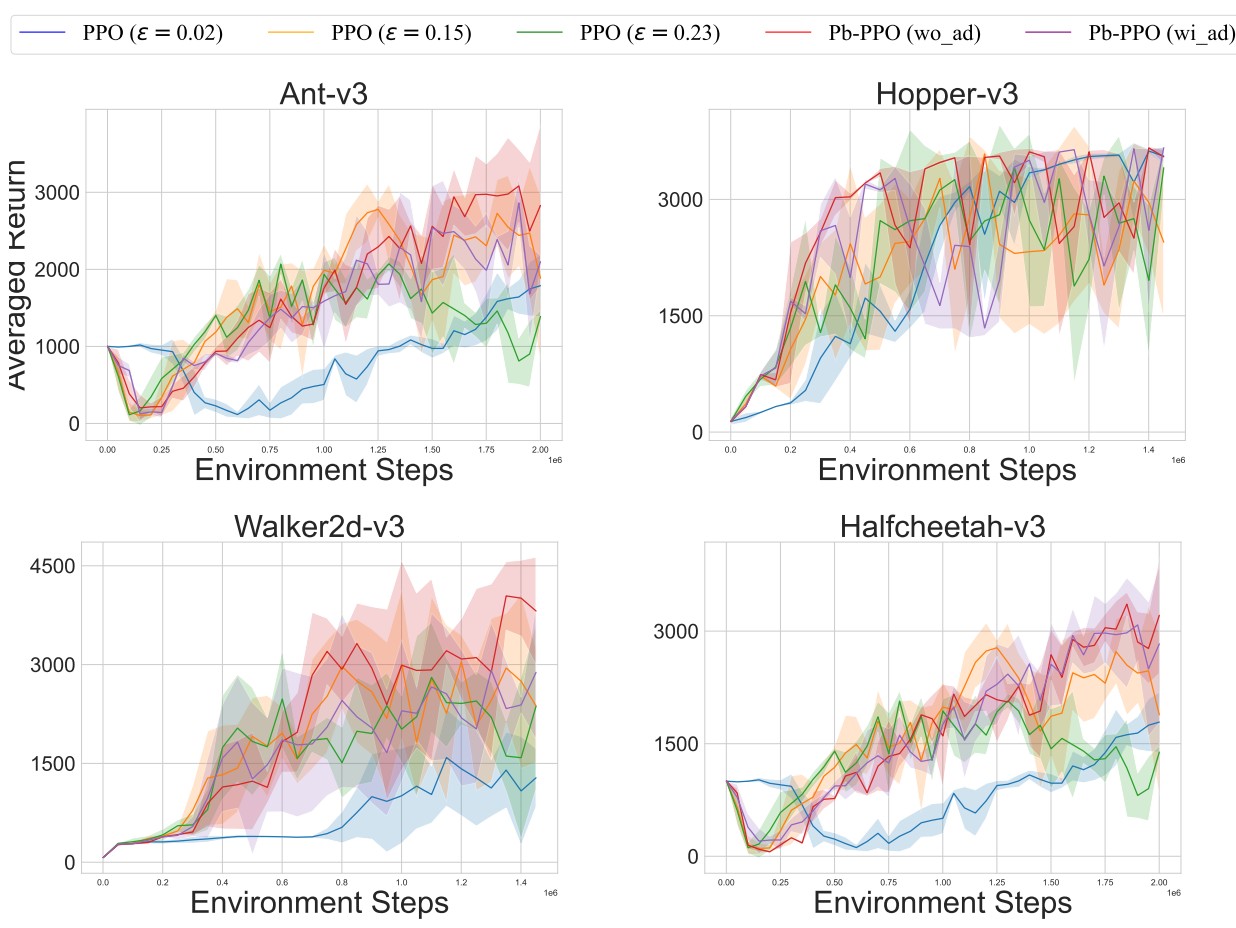

Figure 1: Pb-PPO (task feedback) on locomotion tasks. Each solid curve in these figures represent the average experimental results across multiple seeds, and the shadowed area corresponds to the fluctuation of Return curves.

Table 1: Performance comparision of the last training outcomes. We average the last training results of PPO with diverse clipping bounds and Pb-PPO across multiple seeds, we also compare with the experimental results of TRPO, DDPG, official PPO, TRGPPO, etc. are directly quoted from (Fujimoto et al., 2018), and PPO-$\lambda$ is quoted from (Chen et al., 2018).

| Task | PPO ($\epsilon$=0.02) | PPO ($\epsilon$=0.15) | PPO ($\epsilon$=0.23) | PPO (official) | TRPO | DDPG | PPO-$\lambda$ | TRGPPO | Pb-PPO |
|---|---|---|---|---|---|---|---|---|---|
| Ant-v3 | 1686±276 | 2458±476 | 2457.9±506.6 | 1083.2 | -75.85 | 1005.3 | - | - | **3151.8±545.9** |
| Halfcheetah-v3 | 1284±81 | 1608±68 | 1568±65 | 1795.4 | -15.6 | 3305.6 | - | 4986.1 | 2781.4±1188.0 |
| Hopper-v3 | 2858±275 | 2274±591 | 2327±523 | 2164.7 | 2471.3 | 1843.9 | 1762.3 | 3200.5 | **3414.9±216.9** |
| Walker2d-v3 | 1887±229 | 2542±304 | 2340±304 | 3317.7 | 2321.5 | 1843.9 | 2312.1 | 3886.8 | **3913.1±699.4** |
| **Avg.** | 1623.8 | 1869.6 | 1836.4 | 2090.3 | 1175.3 | 1999.7 | - | - | 3315.3 |

Pb-PPO's performance on robotic tasks, we test it in legged-gym and navigation tasks. Specifically, our Gym-Mujoco benchmarks include various of locomotion tasks such as `Walker2d-V3`, `Hopper-V3`, `HalfCheetah-V3`, and `Ant-V3`. For the legged-gym tasks Rudin et al. (2022), we evaluated a quadruped ( The robot tag is `anymal-c`) robot's performance on both flat and complex terrains (*In the legged gym environment, the training objective is a goal-conditioned policy, meaning that a policy is given a goal and makes decisions based on this goal.*). The navigation tasks are designed based on the ROS-Gazebo [2] environment for autonomous vehicles. In this environment, the vehicle needs to navigate around various obstacles and reach three predefined

---

[2]https://gazebosim.org/docs/garden/ros_installation

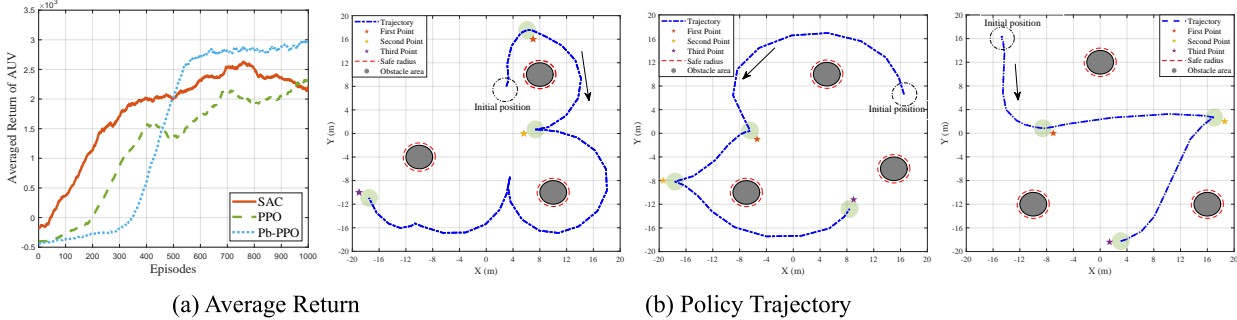

(a) Average Return                    (b) Policy Trajectory

Figure 2: **Pb-PPO on AUV navigation tasks across different difficulty levels.** (a) Average Return curve in the hard navigation task. (b) From left to right are trajectories of the AUV in the easy, medium, hard environment in turn, we introduce the environment setting in Appendix.

targets sequentially. Based on the arrangement of obstacles and targets, the navigation environment can be divided into three levels: `easy`, `medium`, and `hard`, meanwhile, detailed environment settings can be found in Appendix.

**Baselines.** Our baseline models primarily consist of PPO variants with different clipping bounds. Additionally, we compare the final training results of Pb-PPO with Trust Region Policy Optimization (TRPO) (Schulman et al., 2017a), and various on-policy algorithms including Deep Deterministic Policy Gradient (DDPG) (Lillicrap et al., 2019), Trust Region-Guided PPO (TRGPPO) (Wang et al., 2019), and PPO-$\lambda$ (Chen et al., 2018).

## 5.1 Experimental Results on Gym-Mujoco domain.

Our main experimental results are illustrated in Figure 1 and table 1. Overall, both settings of Pb-PPO demonstrate favorable performance in the selected locomotion tasks. Specifically, in our chosen locomotion tasks, Pb-PPO exhibits a advantage in the `Walker2d`, `HalfCheetah`, and `Ant` tasks, and it also demonstrates faster convergence in the `Hopper` tasks. Meanwhile, we summarize the final training results of PPO baseline and various on-policy algorithms in Table 1, where Pb-PPO outperforms the most of selected on-policy algorithms. Apart from its advantages in algorithm performance and convergence speed, Pb-PPO also shows consistently stable training curves across various locomotion tasks (without significant gaps). Our main experimental results are illustrated in Figure 1 and table 1. Overall, both settings of Pb-PPO demonstrate favorable performance in the selected locomotion tasks. Specifically, in our chosen locomotion tasks, Pb-PPO exhibits a advantage in the `Walker2d`, `HalfCheetah`, and `Ant` tasks, and it also demonstrates faster convergence in the `Hopper` tasks. Meanwhile, we summarize the final training results of PPO baseline and various on-policy algorithms in Table 1, where Pb-PPO outperforms the most of selected on-policy algorithms. Apart from its advantages in algorithm performance and convergence speed, Pb-PPO also shows consistently stable training curves across various locomotion tasks (without significant gaps).

## 5.2 Pb-PPO on robotic tasks

We further test Pb-PPO's performance in multiple simulation environments designed for real-world deployment, including autonomous vehicle and legged gym tasks.

**Pb-PPO can complete navigation tasks of varying difficulty levels.** We tested Pb-PPO in a simulated environment for obstacle avoidance and navigation tasks with an autonomous vehicle. As shown in Figure 2 (a), Pb-PPO has better Return curve that surpass the majority online algorithms including PPO and Soft Actor Critic (SAC) Haarnoja et al. (2018) on the hard navigation task. Furthermore, as shown in Figure 2 (b), the policies trained with Pb-PPO can consistently navigate around all obstacles and reach the preset targets in nearly the shortest path, demonstrating Pb-PPO's stable evaluation performance in

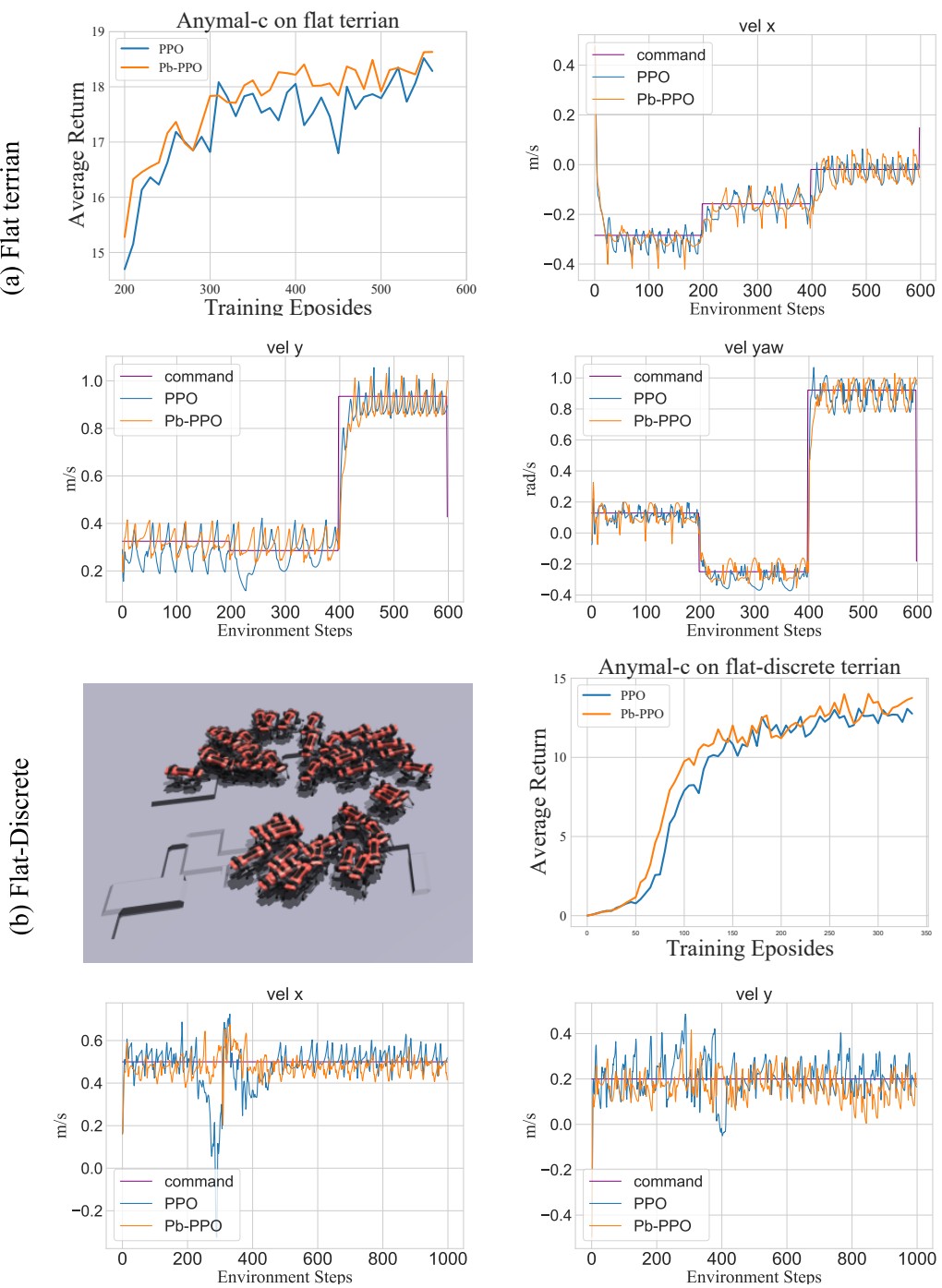

Figure 3: (a) Pb-PPO on flat terrain: The first plot shows training curves (average returns in multi-robot parallel training). Subsequent plots display x/y linear velocities and angular velocity during evaluation with 50 robots initialized using the trained policy. (b) Pb-PPO on complex terrain: We evaluated the complex-terrain-trained policy in flat-discrete terrain, with remaining plots following the same format as (a).

multi-target obstacle avoidance tasks. (*Additionally, we supplement Pb-PPO in underwater obstacle avoidance tasks in the Appendix, where Pb-PPO still exhibits stable performance.*)

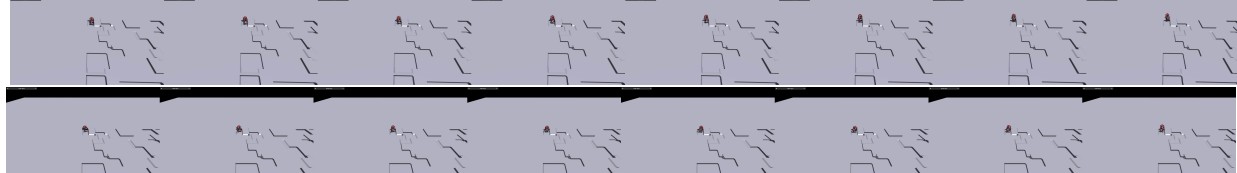

Figure 4: The walking states of quadruped robots (`anymal-c`) on complex terrains. The upper figure shows the quadruped robot trained by Pb-PPO, and the lower figure shows the quadruped robot trained by PPO. In general, Pb-PPO exhibits a more stable gait when climbing stairs.

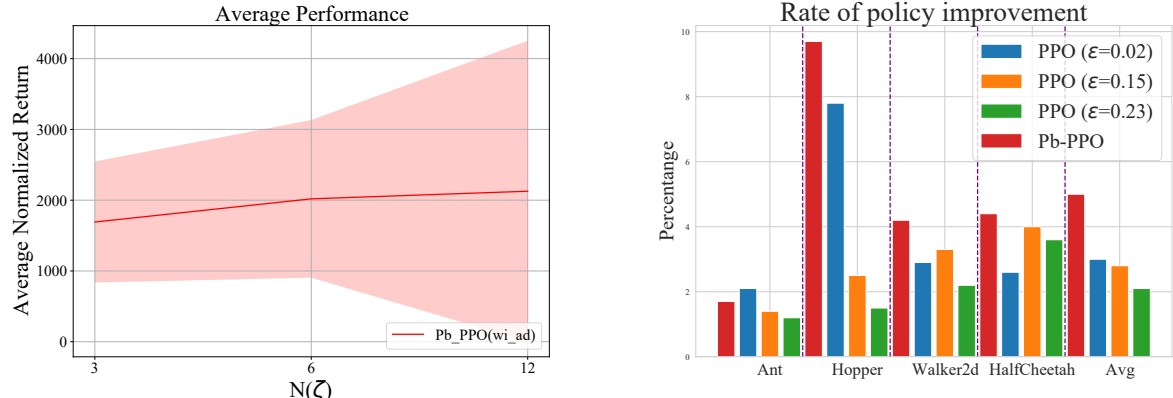

Figure 5: (a) Return of Pb-PPO across different num($\zeta$). (b) The success rate of policy improvement.

**Pb-PPO demonstrates more stable performance on quadruped robots compared to PPO.** We also evaluate Pb-PPO on quadruped robots in the legged-gym environment. As shown in Figure 3(a), we train and test Pb-PPO on flat terrain in parallel. Compared to PPO, Pb-PPO demonstrate consistently better training curves. Meanwhile, we conducted parallel tests with 50 quadruped robots on flat terrain and found that Pb-PPO's policy more closely adhered to the given commands, thereby validating its superior stability. Furthermore, as illustrated in Figure 3(b), we used pre-trained policies on discrete, flat 1:1 terrain and initialized 50 robots to test on flat-discrete mixed terrain. Pb-PPO exhibits a better Return curve, and stronger response to commands than PPO. We believe the advantage of Pb-PPO is due to its ability to dynamically adjust the clipping bound based on task feedback, achieving a stable balance between model updates and exploration. This advantage is reflected not only in the Return curves but also in the stability of the test tasks. However, in these quadruped robot tasks, the superiority of Pb-PPO over PPO in terms of Return curves was less pronounced compared to single-environment tests (Gym locomotion, navigation). We attribute this to the reward in parallel environments reflecting the overall task performance rather than setting the most appropriate bound for each policy with fine granularity. Therefore, Pb-PPO's performance in parallel environments should have to be further improved.

**Actual motion states of policies trained by Pb-PPO.** Currently, in Figure 3, we have not shown the motion states of more agents. Therefore, we further supplement more motion states in Figure 4. As shown in Figure 4, the policy trained by Pb-PPO exhibits higher stability when climbing stairs compared to PPO. Additionally, we have provided videos in the supplementary materials for reviewers to conduct systematic evaluations.

## 6 Ablation

**Variations in the performance of Pb-PPO as influenced by different numbers of bandit arms num($\zeta$).** In this section, we further validate the effectiveness of our approach by changing the number of bandit arms. Illustrated in Figure 5, the training performance, averaged across `Walker2d`, `Hopper`, and `Humanoid` for Pb-PPO (wi-ad), exhibits sensitivity to the number of bandits. Notably, the performance of Pb-PPO demonstrates improvement with an increasing num($\zeta$) (across 3, 6, 12).

**Rendering the success rate of Pb-PPO's policy improvement.** We count the times that new policy is better than old policy as $N_{\text{success}}$, and compute the success ratio by dividing the whole iterations $N$, *i.e.* $\frac{N_{\text{success}}}{N}$. If $\frac{N_{\text{success}}(\text{Pb-PPO})}{N(\text{Pb-PPO})}$ is higher than $\frac{N_{\text{success}}(\text{PPO})}{N(\text{PPO})}$, the feasibility of Pb-PPO can be further validated. As shown in Figure 5, we average the training results across `Hopper`, `Walker2d`, `Ant`, `HalfCheetah`, our Pb-PPO achives 5.0% success rate which is the best across all selected baselines.

## 7 Conclusion and Limitation

In this study, we propose Pb-PPO which utilizing bandit algorithm to dynamically adjust the clipping bound during the process of proximal policy optimization. Pb-PPO showcase better performance than PPO and PPO's variants, even can solve all tasks with single set of hyper-parameters. However, Pb-PPO's performance in parallel environments should have to be further improved. In the future, we will scale our approach to the setting reflecting human preference.

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

## A   Mathematic Proof of Equation 5

According to Hoeffding's inequality, we know that:

$$|\frac{R_{\epsilon_i} - \mathbb{E}[R|\epsilon_i]}{n}| \leq \frac{R_{\epsilon_i}^{\max} - R_{\epsilon_i}^{\min}}{n}\sqrt{\frac{1}{2}\ln\frac{2}{\sigma}}$$

$$\rightarrow R_{\epsilon_i} - \mathbb{E}[R|\epsilon_i] \leq (R_{\epsilon_i}^{\max} - R_{\epsilon_i}^{\min})\sqrt{\frac{1}{2}\ln\frac{2}{\sigma}} \tag{9}$$

$$\rightarrow R_{\epsilon_i} \leq \mathbb{E}[R|\epsilon_i] + (R_{\epsilon_i}^{\max} - R_{\epsilon_i}^{\min})\sqrt{\frac{1}{2}\ln\frac{2}{\sigma}},$$

## B   Social Impact

In this research, we propose Pb-PPO, an enhanced version of PPO that achieves superior performance on robotic tasks. Additionally, Pb-PPO demonstrates stable evaluation results, making it promising for fields requiring consistent training. In the future, we plan to explore how to scale Pb-PPO to domains that reflect human preferences.

## C   Computing Resources

We run each task multiple times. Our experiment are running on computing clusters with $16\times4$ core cpu (Intel(R) Xeon(R) CPU E5-2637 v4 @ 3.50GHz), and $16\times$RTX2080 Ti GPUs

## D   Scaled Experiments

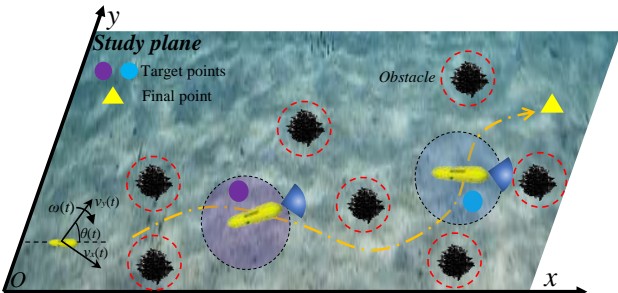

Figure 6: Policy trajectory in AUV navigation system.

**Underwater AUV simulation setting.**   As shwon in Figure 6, the simulation is carried out on a 40m×40m area with a water depth of -200m, on which the obstacles are randomly distributed. At the beginning, the position of the AUV is randomly distributed, and the AUV knows its own position. The area boundaries act as obstacles to restrict the AUVs in the specified area.

## E   Environments and hyper-parameters

In the main context, we test Pb-PPO on costumed AUV navigation environment and legged-gym these two environments. Since AUV navigation environment is designed by our-self, we carefully detailed the modeling of this environment. In terms of legged-gym our training configuration is the same as the original set up of legged-gym, we additionally provide the command during the evaluation process. Our environment codebase is implemented based on this project (related to ROS-Gazebo): `https://gitee.com/fangxiaosheng666/PPO-SAC-DQN-DDPG`

**AUV Navigation.**   We describe the AUV navigation problem as a Markov decision process (MDP), which can be defined by a quintuple, i.e.

$$\mathcal{M} = (\mathcal{S}, \mathcal{A}, d_{\mathcal{M}}(\cdot \mid \mathbf{s}_t, \mathbf{a}_t), \mathcal{R}, \gamma), \tag{10}$$

where $\mathcal{S}$ and $\mathcal{A}$ denotes the state and action space of the AUV, respectively. Moreover, $\gamma$ is the discount factor, while $d_{\mathcal{M}}(\cdot \mid \mathbf{s}_t, \mathbf{a}_t)$ represents the state transition probability function. To be intuitive, at time $t$, AUV selects the

action $\mathbf{a}_t \in \mathcal{A}$ according to its policy $\pi_\theta$ by observing the current state $\mathbf{s}_t \in \mathcal{S}$, and transitions to the next state $\mathbf{s}_{t+1} \sim d_\mathcal{M}(\cdot \mid \mathbf{s}_t, \mathbf{a}_t)$ and gets the reward $r(t) \in \mathcal{R}$. The details are as follows:

**State space:** In the navigation task, the observation space of AUV at time $t$ is $\mathbf{s}_t$, which can be defined as

$$\mathbf{s}_t = [l(t), l_{A \leftrightarrow T}(t), \theta(t), \phi_{A \leftrightarrow T}(t), \chi(t)], \tag{11}$$

where $l(t)$ contains the distances detected by sonar between the AUV and various obstacles, while $l_{A \leftrightarrow T}(t)$ represents the distance between the AUV and the target point. $\theta(t)$ and $\phi_{A \leftrightarrow T}(t)$ respectively indicate the orientation angle (yaw angle) of the AUV and the angle between the AUV and the target point. Furthermore, $\chi(t) \in \{0, 1\}$, and $\chi(t) = 1$ indicates the current training episode has concluded, while vice versa.

**Action space:** In the process of navigation task, the AUV makes action $\mathbf{a}_t$ at time $t$ by observing the state $\mathbf{s}_t$ and action $\mathbf{a}_t$, which can be given by

$$\mathbf{a}_t = [v(t), \omega \mathbf{a}_t], \tag{12}$$

where $\|v(t)\| = \sqrt{v_x(t)^2 + v_y(t)^2}$ and $\|\omega \mathbf{a}_t\|$ indicate the linear and angular velocity of the AUV, respectively. And the AUV can adjust its own motion state by changing its linear and angular velocity.

**Reward function:** We need to design the corresponding reward function to guide the AUV to make reasonable decisions in the complex environment to optimize the navigation trajectory to safely complete the navigation task. The rewards received by the AUV at time $t$ consist of the following parts

$$r_c(t) = -500 \operatorname{ceil}(l_{\text{safe}} / \min(l(t))), \tag{13}$$

$$r_g(t) = 1000 \operatorname{ceil}(l_{A \leftrightarrow T}^{\max} / l_{A \leftrightarrow T}(t)), \tag{14}$$

$$r_e(t) = -0.2 + 5(l_{A \leftrightarrow T}(t-1) - l_{A \leftrightarrow T}(t)) + 2(\phi_{A \leftrightarrow T}(t-1) - \phi_{A \leftrightarrow T}(t)), \tag{15}$$

where $r_c(t)$ is a penalty term used to prevent the AUV from colliding with the obstacles, while $l_{\text{safe}}$ is the safe distance between the AUV and the obstacles, and $\operatorname{ceil}(x)$ is the binary function, which means that $\operatorname{ceil}(x)$ equals to 1 when $x \geq 1$, and equals to 0 when $x \leq 1$. Additionally, when the AUV detects the target point for the first time, it receives a reward $r_g(t)$. In addition, we use the reward item $r_e(t)$ to encourage the AUV to get closer to the target point. Therefore, the total reward available for AUV at time $t$ can be weighted by

$$r(t) = \delta_c r_c(t) + \delta_g r_g(t) + \delta_e r_e(t), \tag{16}$$

where $\delta_c$, $\delta_g$ and $\delta_e$ are the weights of each reward or penalty item, respectively, which can be adjusted according to the application needs.

Based on the above analysis, we summarize several engineering constraints that need to be considered during the actual navigation process, and formulate a constraint optimization problem whose goal is to optimize the policy of the AUV to maximize the total expected return. The constrained optimization problem can be expressed as

$$\max_{\pi_\theta} J(\theta) = \max_{\pi_\theta} \mathbb{E} \left[ \sum_{t'=t}^{T=\infty} \gamma^{t'-t} r_{t'}(s(t), \pi_\theta(a(t) \mid s(t))) \right], \tag{17a}$$

$$s.t. \ \min(l(t)) \geq l_{\text{safe}}, \tag{17b}$$

$$s.t. \ l_{A \leftrightarrow T}(t) \leq l_{A \leftrightarrow T}^{\max}, \tag{17c}$$

$$v_{\min} \leq \|v(t)\| \leq v_{\max}, \omega_{\min} \leq \|\omega \mathbf{a}_t\| \leq \omega_{\max}, \tag{17d}$$

where Equation 17a denotes the optimization objective, and Inequality 17b represents the constraint that prevents the AUV from colliding with obstacles. Moreover, Inequality 17c stands for the constraint that ensures the AUV to get to the target point, while Inequality 17d restricts the velocity and angular velocity range of the AUV.

Table 2: Command setup when conducting evaluation.

| Physical standard | range |
|---|---|
| x-axis v | [-1.0 m/s, 1.0 m/s] |
| y-axis v | [-1.0 m/s, 1.0 m/s] |
| angular v | [-0.01 rad/s, 0.01 rad/s] |
| heading | [-3.14, 3.14] |

**Legged Gym.** We utilize the default configuration of `"anymal-c"` in legged-gym for policy training and evaluation. In particular, the default configuration of `"anymal-c"` can be referred to `https://github.com/leggedrobotics/legged_gym/tree/master/legged_gym/envs/anymal_c`, which includes plat and complex terrain these two different kinds of settings. Additionally, our complex terrain are composed of 1:1 flat and discrete sub-terrains. During the evaluation stage, we set up the command as shown in table 2, while maintaining the original parameters as: `https://github.com/leggedrobotics/legged_gym/blob/master/legged_gym/scripts/play.py`

