# OpenReview forum: "A Dynamical Clipping Approach with Task Feedback for Proximal Policy Optimization"
_TMLR — Withdrawn by Authors_

### Review · Reviewer_Pqwr · 2025-06-13

**Summary Of Contributions:**

This paper proposes Pb-PPO, a variant of Proximal Policy Optimization that frames the task-dependent clipping bound selection as a multi-armed bandit problem. During training, the clipping ratio is sampled using an upper confidence bound (UCB) strategy from a pre-defined range. The method is evaluated on several locomotion and navigation tasks, showing improved performance compared to standard PPO with fixed clipping bound.

**Audience:**

Yes

**Claims And Evidence:**

Yes

**Requested Changes:**

- As mentioned above, maybe it is necessary to compare with a few other standard hyper-parameter tuning methods.
- The authors did not explain the candidate range of clipping ratios and other related settings, and the paper does not analyze how the initial range affects final performance. An ablation study would be helpful.
- Pb-PPO (wo-ad) appears to outperform the normalized version in most tasks (especially in Walker and Ant). Can the authors explain whether normalization is helpful for your method?

**Strengths And Weaknesses:**

### Strengths
- The paper conducts extensive experiments across multiple domains, for example, locomotion and navigation.

### Weaknesses
- The clipping ratio is selected at the task level and not conditioned on states or dynamics. As such, Pb-PPO can be seen as a form of automatic hyper-parameter tuning. Therefore, maybe it would strengthen the contribution to compare it with a few other standard hyper-parameter tuning methods.
- Some baselines included in the later experiments (e.g., SAC) seem less relevant to the method, while more appropriate PPO variants (e.g., PPO-λ, TRGPPO which the authors mentioned) are omitted in those settings.
- While the authors suggest that their clipping strategy could be combined with other modifications of PPO (e.g., Tao et al., Xu et al., Song et al.) for further improvement, it would be great to provide some empirical validation.
- The improvements over PPO in locomotion tasks seem marginal. In Figure 1, the training curves of Pb-PPO are not clearly better than PPO ($\epsilon$ = 0.15). However, in Table 1 the numbers look significantly better. How do the authors compute the numbers in Table 1? Are they evaluated on a sufficient number of trajectories or averaged across a sufficient number of consecutive training steps or something else?

Typos:
- Abstract: refelect → reflect
- Section 4: “In the problem setting of …” should be “in the problem setting of …”

---

### Review · Reviewer_PAq9 · 2025-06-20

**Summary Of Contributions:**

This paper mainly proposes a variant of PPO based on following:
1. using a multi-armed bandit framwork to dynamically adjust the clipping bound
2. using upper confidence bound to address exploration-exploitation trade off.
Overall, compare with PPO and some of the vaiants, the Pb-PPO should demonstrates more stable training performance and achieves better training results.

**Audience:**

Yes

**Broader Impact Concerns:**

N.A

**Claims And Evidence:**

No

**Requested Changes:**

1. To better see the performance difference, auther may add more benchmarks espicially more complex one such as humanoid.

2. For example, in algorithm box "the counter $N^{bandit}$ of total visitations" and above Equation (7) " total sampling times $N^{bandit}$ " are they meaning the same thing? or different? Since this paper has a lot parameters, I suggest to use the same language across the paper to avoid confusion.

3. Some typo "clpping bound" should be "clipping",  "hyperparameter" not "hyper-parameters", etc

**Strengths And Weaknesses:**

> Strengths:
1. The idea of using multi-armed bandit, bi-level formulation are novel
2. The thereitial proof enhance the soundness of the method
3. The ablation study investigates sensitivity to number of bandit arms.


>Weaknesses:
1. The method treats each candidate clipping bound as an independent arm, ignoring possible correlations between clipping bounds.
2. the major concerns are the benchmark results, from Figure 1, it is really hard to get the claim that Pb-PPO is outperform other PPO variants, espicially for Ant and Hopper environments. Additionally, for command following Figure 3, the performance is very close to make a claim. Btw, auther need to adjust Figure 1 size to fully show the y axis label.
3. Auther make claim "even can solve all tasks with single set of hyper-parameters." not sure which set of hyperparameter is refering to?
4. From the computation side, the added layer of clip-bound selection (especially with many candidate bounds) introduces computational overhead. As Figure 1 and 2 Pb-PPO is learning slowly at beginning.
5. notation and terminology need to be more clear. Details are in Requested changes.

---

### Review · Reviewer_P5nS · 2025-07-02

**Summary Of Contributions:**

The paper proposes Preference-based Proximal Policy Optimization (Pb-PPO), which treats the choice of PPO’s clipping bound as a multi-armed-bandit problem whose “reward” is the episodic return; Subsequently, the author propose a bi-level view of PPO (outer level: choose the bound; inner level: optimize the policy) while adding only one hyper-parameter. Experiments on four Gym-Mujoco locomotion tasks, legged-gym quadruped control, and ROS-Gazebo AUV navigation show faster learning and higher final returns than fixed-clip PPO and prior adaptive-clip variants.

**Audience:**

Yes

**Claims And Evidence:**

No

**Requested Changes:**

- Please provide a formal theoretical result, proposition or at least a lemma showing that the UCB selection rule used in alg. 1 is no-regret w.r.t. the optimal static clip bound.
- Please provide the insight of solving bi-level problem. For example, bilevel gradient method,  regret bound or lower-level optimality conditions

**Strengths And Weaknesses:**

Strength:
-  fixed clipping thresholds in PPO is a well known problem, which has been repeatedly reported to comprise the stability and exploration. Therefore, the topics is interesting and important
- The simplicity of the core idea makes the implementation straightforward, only a light wrapper around PPO. The hyper parameter $\lambda$ is also lightweight.

Limitations:
- While PPO-lambda, TRGPPO, AC-PPO, etc similar works are mentioned, conceptually speaking, the proposed method is really close to these prior “adaptive-clip” works, it limits the novelty contribution
- There is no guarantee that UCB rule would improve the outer objective, i.e. $\max_{\epsilon}J(\pi, \epsilon)$
- The bi level problem is proposed, whereas I expect the author show how the upper and lower level objectives are jointly solved, or at least provide a provably convergence algorithm. However, the authors directly jump to a heuristic, i.e. Equation 7, which is only an ad-hoc departure from the standard UCB
- The advantages of proposed method on robot locomotion are modest and not always statistically significant

---

### Note · Authors · 2025-07-27

**Comment:**

Dear Action Editor and Reviewers,

I hope this email finds you well! I am writing to inform you that, unfortunately, due to ongoing health issues and after receiving medical treatment, I am still not showing significant improvement. As a result, I find myself unable to continue with the necessary work to address the comments.

After careful consideration, I regretfully request to withdraw my manuscript from the review process. I deeply apologize for any inconvenience this may cause to you, the reviewers, and the editorial team. Please understand that this decision comes under circumstances beyond my control, and I sincerely appreciate your understanding and support during this challenging time.

I would like to express my heartfelt gratitude to the AE and reviewers for their time and effort in reviewing my work. I regret that I am unable to fully address their valuable feedback, but I hope that I will have the opportunity to submit my work for consideration again once my health permits.

Thank you for your time, understanding, and continued support. I look forward to hearing from you regarding the withdrawal process.

Sincerely,
Authors

**Withdrawal Confirmation:**

I have read and agree with the venue's withdrawal policy on behalf of myself and my co-authors.